# Robust Contextual Pricing

**Anupam Gupta**
NYU and Google

**Guru Guruganesh**
Google

**Renato Paes Leme**
Google

**Jon Schneider**
Google

## Abstract

We provide an algorithm with regret $O(Cd \log \log T)$ for contextual pricing with $C$ corrupted rounds, improving over the previous bound of $O(d^3 C \log^2(T))$ of Krishnamurthy et al. (2020). The result is based on a reduction that calls the uncorrupted algorithm as a black-box, unlike the previous approach that modifies the inner workings of the uncorrupted algorithm. As a result, it leads to a conceptually simpler algorithm.

Finally, we provide a lower bound ruling out a $O(C + d \log \log T)$ algorithm. This shows that robustifying contextual pricing is harder than robustifying contextual search with $\epsilon$-ball losses, for which it is possible to design algorithms where corruptions add only an extra additive term $C$ to the regret.

## 1 Introduction

In the *contextual pricing* problem, the seller is a learner who, in each round, gets a different product described by a feature vector $x_t$, for which it then posts a price $p_t$. Typically, the buyer purchases the product if its valuation (which is a function $v_t(x_t)$ of the feature vector, and a priori unknown to the seller) is above this price. All the seller observes, after posting the price, is whether the buyer purchases the product or not. The goal of the learner (i.e., seller) is to maximize their revenue $\sum_t p_t \mathbf{1}(\text{item } t \text{ sold})$. As usual in pricing problems, the learner faces the exploration/exploitation trade-off: it can "explore" by setting aggressive prices that allow it to learn about the buyer's valuation, but then it risks losing the sale; or it can "exploit" by setting conservative prices that guarantee a sale, but potentially get a suboptimal revenue. In the revenue management literature, this is referred to as the *learn-and-earn* problem.

Such pricing problems pose two main challenges:

(i) we only observe binary feedback (*sale* or *no-sale*), but do not directly observe the loss in each round;

(ii) the loss function is *asymmetric* and *discontinuous*: while under-pricing an item by an amount $x$ will result in a sale with loss $x$ with respect to the best possible revenue, over-pricing will cause the buyer not to purchase the produce and incur a large constant loss.

For the single-dimensional non-contextual version of this problem, the seminal paper of Kleinberg and Leighton (2003) identifies the best way to handle this asymmetry, obtaining the optimal regret of $\Theta(\log \log T)$. Since then, the Kleinberg and Leighton (2003) paper has been extended to the $d$-dimensional contextual case by a series of papers that have obtained increasingly better bounds: the work of Cohen et al. (2016) gave $O(d^2 \log T)$, Leme and Schneider (2018) showed $O(d^4 \log \log T)$, and finally Liu et al. (2021) gave $O(d \log \log T)$, matching the lower bound.

So far, our discussion has described the realizable and noise-free version of the problem: the buyer has a fixed value $v_t$, which is a constant in $[0, 1]$ in the non-contextual case, or a linear function $v_t(x_t) = \langle v^*, x_t \rangle$ in the contextual case. Moreover, the purchasing decision is based on whether the posted price $p_t$ is above or below this value $v_t$. There has been considerable work extending the

model to handle stochastic noise models: see, e.g., Javanmard and Nazerzadeh (2016); Cohen et al. (2016); Javanmard (2017); Shah et al. (2019); Liu et al. (2021); Xu and Wang (2021, 2022).

Comparatively, adversarial noise models (also known as *corrupted feedback*) are much less understood. The study of corrupted feedback for contextual pricing was pioneered by Krishnamurthy et al. (2020). In this model, there is still a ground truth value $v_t$ and the buyer again makes most of the purchasing decisions based on whether $p_t \leq v_t$. However, for a certain number of rounds $C$, the decisions are arbitrary. This can model that buyers often make irrational choices (as in Krishnamurthy et al. (2020)) or that an adversary can corrupt the feedback given to the learner (as in Lykouris et al. (2018); Paes Leme et al. (2022)).

## 1.1 Our Results

We give improved algorithms and lower bounds for the contextual pricing problem in settings with corrupted feedback.

**Our Results: Upper Bounds.** Our paper proposes a reduction from corrupted to uncorrupted case in contextual pricing and through that, obtains conceptually simpler algorithms with improved regret. Our results are the following:

1. For the *known corruption setting*, We give an algorithm that achieves a regret of

$$O((C+1) \cdot d \log \log T).$$

   This improves over the $O(Cd^3 \log^2 T)$ bound of Krishnamurthy et al. (2020).

2. For the setting of *unknown corruption $C$ with a known upper bound $U$* (i.e., we are promised that the actual number of corruptions $C$ is at most $U$), we give an algorithm with regret

$$O((C + \log U) \cdot d \log \log T).$$

   This improves over the regret bound of $O((C + \log U)d^3 \log^2 T)$ given by Krishnamurthy et al. (2020).

At a high level, our approach is to execute several parallel copies of an uncorrupted contextual pricing algorithm and query each of them in each round for advice on which price to set, choosing the maximum of such prices to post. In the case of a sale, we choose a random algorithm in the pool to update. We set the update probabilities in a way that one of the parallel copies will never observe a corruption with high probability. The final step is to bound the regret of the entire algorithm by comparing it with the regret of the instance that never observes a corruption.

The main observation is to exploit the following property of the pricing loss: if we choose the maximum price and it results in a sale, the loss incurred it at most the loss of *any* given algorithm. This allows us to charge the loss of the multiple algorithms maintained by our algorithm to the loss of the benchmark, i.e., the single optimal algorithm that observes uncorrupted feedback.

**Our Results: Lower Bounds.** The lower bound on the regret for the uncorrupted case is $\Omega(d \log \log T)$, so the best regret bound one can hope for is $O(C + d \log \log T)$. For the related problem of contextual search with $\epsilon$-ball loss (i.e., with a different loss function), Paes Leme et al. (2022) show a regret bound of $O(C + d \log(1/\epsilon))$ which adds only an additive $C$ overhead with respect to the tight bound of $O(d \log(1/\epsilon))$ for the uncorrupted case. The natural question is: can we hope to achieve algorithms with regret $O(C + d \log \log T)$?

In Theorem 5.2 we show this is not possible: we give a lower bound ruling out algorithms with regret $O(C + d \log \log T)$, and hence showing that robustifying contextual pricing is harder than robustifying other contextual search problems (e.g., for the $\epsilon$-ball loss function).

These lower bounds also inspire new algorithmic results: using the ideas in the lower bound, we are able to provide in Theorem 6.1 a $O(C + \log T)$-regret algorithm for the *corrupted non-contextual case* when corruptions are one-sided: i.e., the adversary is allowed to corrupt no-sales to sales but not the other way round.

## 1.2 Related Work

Our paper lies in the intersection of the literature of learning to price, contextual search, and learning with corrupted feedback. Learning to price was initiated by Kleinberg and Leighton (2003) who provide matching upper and lower bounds for the regret achievable by a seller posting prices to a buyer under different assumptions on the buyer's valuation. This setting was later extended to auctions Mohri and Medina (2014); Cesa-Bianchi et al. (2014) and strategic agents Amin et al. (2013); Drutsa (2017).

Contextual pricing is a generalization of learning to price to higher dimensional settings, which is a part of a larger class of problems known as contextual search which explore binary feedback in higher dimensional settings for various loss functions. A long line of paper starting with Amin et al. (2014) provide increasingly better bounds for contextual search with the pricing loss and other types of losses: Cohen et al. (2016); Lobel et al. (2018); Leme and Schneider (2018); Liu et al. (2021).

Finally, learning with corruptions correspond to the study of settings in which the data follows a predictable pattern with the exception of a few examples whose label is arbitrary. A very early example is called Ulam's game (Ulam (1976)) in which a learner tried to identify a number on a sorted list using comparison queries but $C$ of those queries can have erroneous answers. Upper and lower bounds to this setting were given by Rivest et al. (1978).

There has been recent interest in providing corruption robust bounds to contextual search problems. This was pioneered by Krishnamurthy et al. (2020) who study contextual pricing in the presence of corruptions—the exact same setting as in our paper. Recently, Paes Leme et al. (2022) studies the contextual search problem under the symmetric and $\epsilon$-ball losses with corrupted feedback.

## 2 Model and Definitions

**(Uncorrupted) Contextual Pricing** In the contextual pricing problem, the buyer's valuation is represented by a vector $v^* \in \mathbb{B}_d = \{x \in \mathbb{R}^d; \|x\|_2 \leq 1\}$ that is unknown to the learner. In each round $t = 1 \ldots T$, the learner will be presented an item described by a feature vector $x_t \in \mathbb{B}$ and will have the opportunity to set a price $p_t$ for the buyer. In the standard (un-corrupted) model, the buyer buys if the value $\langle v^*, x_t \rangle \geq p_t$ in which case the learner obtains revenue $p_t$; otherwise, if the value $\langle v^*, x_t \rangle < p_t$ then the buyer doesn't buy, in which case the learner obtains revenue zero. As usual, the learner is faced with the trade-off of proposing aggressive prices to learn more information about $v^*$, and setting conservative prices that will be guaranteed to sell and generate revenue. To summarize, the feedback $\sigma_t \in \{0, 1\}$ and the loss $\ell_t$ are given by:

$$\sigma_t = \mathbf{1}\{v_t \geq p_t\} \qquad \ell_t(p_t; v_t) := \sigma_t p_t - v_t$$

for $v_t = \langle v^*, x_t \rangle$. Liu et al. (2021) recently showed matching upper and lower bounds of $O(d \log \log T)$ on the total loss $\sum_t \ell_t$ (also referred to as the *regret*) which is a contextual version of the algorithm of Kleinberg and Leighton (2003).

**Corrupted Feedback** In the corrupted version of this problem, the learner has access to feedback $\sigma_t \in \{0, 1\}$. The corruption level $C$ is defined to be the number of periods for which the feedback doesn't match $\mathbf{1}\{v_t \geq p_t\}$, i.e.,

$$C = |\{t \in [T]; \sigma_t \neq \mathbf{1}\{v_t \geq p_t\}\}|$$

In the known corruption case, the algorithm is given access to $C$. In the unknown corruption case, the algorithm knows an upper bound $U$ on $C$ (e.g., $U = T$ in the worst case), and the regret bounds can depend on both $C$ (the actual corruption), and on $U$ (the upper bound, ideally logarithmically or better).

Note that we will still use the term *regret* to refer to the total loss $\sum_t \ell_t$ in the corrupted setting (with the sum ranging over all rounds $t$, even those with corrupted feedback). This deviates slightly from the definition of regret employed in Krishnamurthy et al. (2020), which only counts losses in uncorrupted rounds – note, however, that both notions of regret differ by at most $C$ and hence do not affect any of the obtained asymptotic regret bounds (which even at best have some additive dependence on $C$).

# 3 Reduction in the Known-Corruption Setting

We will present a reduction from the uncorrupted setting to the corrupted setting for contextual pricing. In this reduction, we will treat the uncorrupted algorithm as a black-box described by two subroutines: $\mathsf{Query}(x_t)$ which returns a price $p_t$ and $\mathsf{Update}(x_t, p_t, \sigma_t)$. Given those subroutines, the uncorrupted algorithm can be described as follows: (i) $\mathsf{Query}(x_t)$ returns a price $p_t$ and doesn't modify the internal state of the algorithm; (ii) the learner posts price $p_t$ and observes feedback $\sigma_t$; (iii) $\mathsf{Update}(x_t, p_t, \sigma_t)$ updates the state of the algorithm.

Given an algorithm defined by $\mathsf{Query}(x_t)$ and $\mathsf{Update}(x_t, p_t, \sigma_t)$ and $q \in [0, 1]$ we define its $q$-slowed-down version as the algorithm that prices according to $\mathsf{Query}(x_t)$ but only calls $\mathsf{Update}(x_t, p_t, \sigma_t)$ with probability $q_t \geq q$ upon receiving feedback. With remaining probability the state of the algorithm is unchanged.

**Theorem 3.1.** If there exists an uncorrupted algorithm with loss at most $R(T)$ such that:

1. its suggested price results in at most $R(T)$ no sales

2. the loss of its $q$-slowed-down version is at most $R(T)/q$

then there exists an algorithm with total loss at most $O(C \cdot R(T))$ in the corrupted setting with known corruptions.

**Algorithm Description.** We instantiate $C + 1$ copies of the uncorrupted algorithm denoting the $i$-th copy by $\mathsf{Query}_i(x_t)$ and $\mathsf{Update}_i(x_t, p_t, \sigma_t)$. We also maintain a set of active instances $A$ that is initially $\{1, 2, \ldots, C + 1\}$. Now, for each $t = 1 \ldots T$ we:

- obtain the price proposed by each algorithm $p_{t,i} = \mathsf{Query}_i(x_t)$.
- choose the largest price $p_t = \max_{i \in A} p_{t,i}$ to post.
- observe feedback $\sigma_t$
    - if $\sigma_t = 0$ (no-sale), let $i \in A$ be the index such that $p_t = p_{t,i}$. Then call $\mathsf{Update}_i(x_t, p_t, \sigma_t)$. If the $i$-th algorithm was updated with more than $R(T)$ no sales, then remove $i$ from the active set $A$.
    - if $\sigma_t = 1$ (sale), then let $i$ be a random index from $A$ and call $\mathsf{Update}_i(x_t, p_t, \sigma_t)$.

*Proof.* First we observe that there is at least one index $i^*$ that never observes a corruption, hence the guarantees of the uncorrupted algorithm apply to it. By the first assumption in Theorem 3.1, this algorithm is never eliminated from the the active set.

To bound the total loss, we first observe that the total loss from periods which resulted in a no-sale is at most $(C + 1)R(T)$ since each copy of the algorithm can generate at most $R(T)$ no-sales until it is eliminated.

Now, to bound the total loss from periods in which there was a sale, observe that if the maximum price $p_t$ resulted in a sale, then the price under $i^*$ would also have resulted in a sale since $p_{t,i^*} \leq p_t$ and furthermore, the loss $\ell(p_t; v_t) \leq \ell(p_{t,i^*}; v_t)$. Hence, the expected loss from the periods resulting in sales can be upper bounded by the loss of a $(C + 1)^{-1}$-slowed-down version of the copy $i^*$ of the algorithm which is $(C + 1) \cdot R(T)$. $\qquad\square$

## 3.1 Analysis of the Slowed Down Algorithm

To conclude the proof, we show that the uncorrupted algorithm of Liu et al. (2021) satisfies the conditions of properties in Theorem 3.1. We start by giving an overview of that procedure:

**Summary of the algorithm of Liu et al. (2021)** The algorithm keeps a set $S_t \subseteq \mathbb{B}$ of vectors $v$ consistent with the feedback observed so far. It also keeps $k = \log\log T$ potential functions $\Phi_1(S_t), \ldots, \Phi_k(S_t)$ associated with the doubly-exponential intervals $I_j = [1/2^{2^j}, 1/2^{2^{j-1}}]$ for $j > 1$ and $I_1 = [1/4, 1]$ for $j = 1$. The potential functions depend only on the set $S_t$.

In each round $t$, the algorithm computes the width

$$\mathsf{width}(S_t; x_t) = \max_{v \in S_t} \langle v, x_t \rangle - \min_{v \in S_t} \langle v, x_t \rangle$$

If the width is below $1/T$, the algorithm prices at the lower bound $\min_{v \in S_t} \langle v, x_t \rangle$ guaranteeing a sale with loss at most $1/T$. Otherwise, we choose an index $j$ such that $\mathsf{width}(S_t; x_t) \in I_j$. The algorithm then chooses a price $p_t$ as a function of $j$, $S_t$ and $x_t$ and update the set $S_t$ to

$$S_{t+1} = \{v \in S_t; (2\sigma_t - 1)(\langle v, x_t \rangle - p_t) \geq 0\}$$

- if there is a sale, then the loss is at most $\mathsf{width}(S_t, x_t) = O(1/2^{2^j})$ and the $j$-th potential decreases by $\Phi_j(S_{t+1}) \leq (1 - O(1/2^{2^j})) \cdot \Phi_j(S_{t+1})$. The remaining potentials weakly decrease.
- if there is no sale, then the loss is at most $1$ and the $j$-th potential decreases by $\Phi_j(S_{t+1}) \leq O(1/2^{2^j}) \cdot \Phi_j(S_{t+1})$. The remaining potentials weakly decrease.

Finally, the potentials are such that

$$\Phi_j(S_t) \geq 1/2^{\Omega(d \cdot 2^j)},$$

which bounds the number of possible sales and no-sales for $\mathsf{width}(S_t; x_t) \geq 1/T$. In particular, there can be at most

$$N_j^s := O(d \log d 2^{2^{j-1}} + 2^{j+2^{j-1}})$$

sales and

$$N_j^n := O(d \log d / 2^{j-1} + d)$$

no-sales whenever the width is in the interval $I_j$. This leads to the overall bound of

$$\sum_j N_j^n + (1/2^{2^j}) \cdot N_j^s = O(d \log \log T + d \log d).$$

**Lemma 3.2.** *The uncorrupted algorithm of Liu et al. (2021) satisfies the properties in Theorem 3.1 for $R(T) = O(d \log \log T + d \log d)$.*

*Proof.* The first property is trivially satisfied since no-sales are accounted as a loss of $1$ in the analysis above. Now, to analyze the $q$-slowed down algorithm we can keep the same potentials $\Phi_j(S_t)$ with the difference that they are updated with probability $q$. The time between two updates of each potential is a geometric random variable with probability $q$. So the expected number queries with width in $I_j$ with a sale is $N_j^s/q$, for $N_j^s$ defined above in the algorithm summary. The total loss from sales is therefore at most $(N_j^s/q) \cdot 1/2^{2^j}$ in expectation. The lemma follows from the fact that

$$\sum_j N_j^s \cdot 1/2^{2^j} \leq O(d \log \log T + d \log d)$$

by the analysis of the uncorrupted algorithm. $\qquad\square$

## 4 Unknown-Corruption Setting

Here we assume that we have only an upper bound $U$ on the number of corruptions, i.e., we know that $C \leq U$ but don't know the value of $C$. We provide an algorithm that depends linearly on $C$ (the actual corruptions) but only logarithmically on $U$. The ideas will be to use a version of the multi-layer approach of Lykouris et al. (2018), in which $O(\log U)$ copies of the original uncorrupted algorithm are run in parallel and each copy is updated with exponentially smaller probability. The copies in which the update probability is smaller than $1/C$ are unlikely to experience a corruption.

**Algorithm Description** For a parameter $\delta \in [0, 1]$, we instantiate $k = \log(U/\delta)$ copies of the uncorrupted algorithm denoting the $i$-th copy by $\mathsf{Query}_i(x_t)$ and $\mathsf{Update}_i(x_t, p_t, \sigma_t)$. We also maintain a set of active instances $A$ that is initially $\{1, 2, \dots, k\}$. Now, for each $t = 1 \dots T$ we:

- obtain the price proposed by each algorithm $p_{t,i} = \mathsf{Query}_i(x_t)$.
- choose the largest price $p_t = \max_{i \in A} p_{t,i}$ to post.
- observe feedback $\sigma_t$

- if $\sigma_t = 0$ (no-sale), let $i \in A$ be the index such that $p_t = p_{t,i}$. Then call $\mathsf{Update}_i(x_t, p_t, \sigma_t)$. If the $i$-th algorithm was updated with more than $R(T)$ no sales, then remove all indices smaller or equal to $i$ from the active set $A$.
- if $\sigma_t = 1$ (sale), then choose an index $i$ with probability $1/2^i$ and call $\mathsf{Update}_i(x_t, p_t, \sigma_t)$.

We note that the algorithm only differs from the known-corruption case in the probabilities used to update each algorithm. Also note that with non-zero probability we update algorithm that is no longer active. Such update is clearly useless, but we do so in order to preserve the probability that the still-active algorithms are updated.

**Theorem 4.1.** Given an uncorrupted algorithm wiht regret $R(T)$ following the assumptions of Theorem 3.1 and a known upper bound $U$ on the (unknown) level of corruption $C$, there for every $\delta > 0$, there exists an algorithm that with probability $1 - \delta$ has total loss at most $O((C/\delta) + \log(U/\delta)) \cdot R(T))$.

*Proof.* Let $i^*$ be the index such that $C/(2\delta) \leq 2^{i^*} \leq C/\delta$. For each corrupted round $t$ we update $i^*$ with probability $O(\delta/C)$. Across all the rounds, the probability it experiences no corruption is $(1 - \delta/C)^C = 1 - O(\delta)$. With that probability, this algorithm is never eliminated from the active set $A$ and the bounds from the uncorrupted algorithm will hold.

We again bound the total loss of sales and no-sales separately. Since each algorithm can incur at most $R(T)$ no sales before it is eliminated, the total loss from no-sales is at most $\log(U/\delta) \cdot R(T)$.

To bound the loss from sales, observe that if the maximum price resulted in a sale, then $p_{t,i^*} \leq p_t$ must also result in a sale, so the loss is at most the loss of the loss of a $(\delta/C)$-slowed down version of the uncorrupted algorithm with is at most $(C/\delta) \cdot R(T)$. $\qquad\square$

## 5   Lower Bound

We will study lower bounds for the special case of $d = 1$ in which the valuation is a scalar and is non-contextual. This is the original setting of Kleinberg and Leighton (2003). The lower bound for the uncorrupted pricing problem is $\Omega(\log \log T)$. The best upper bound we can hope for (even with a known number of corruptions $C$) is $O(C + \log \log T)$. The following lemma rules out this bound by a reduction to the lower bound for the *online product design* problem given by Emamjomeh-Zadeh et al. (2021).

**Theorem 5.1.** No deterministic algorithm for the one-dimensional contextual pricing problem with (known) corrupted feedback can guarantee total loss at most $O(C + \log \log T)$. In particular, when $C = \Theta(\log \log T)$, any deterministic algorithm must incur at least $\Omega(C^2/\log C)$ total loss.

We will prove Theorem 5.1 via reduction to the previously studied problem of *online product design* by Emamjomeh-Zadeh et al. (2021). Online product design is a variant of contextual pricing where there are $n$ different possible items, each with an unknown (one-dimensional) true value $v_i \in [0, 1]$. At each round $t \in [T]$, the learner can choose an item $i_t \in [d]$ and set a price $p_t \in [0, 1]$ for this item. If $p_t \leq v_{i_t}$, the item sells at this price and the learner obtains revenue $p_t$, otherwise the item goes unsold and the learner obtains revenue $0$. The learner would like to minimize their regret compared to their optimal strategy, which is to always select the item $i^*$ with the largest value $v_{i^*}$ and price it at its value.

Emamjomeh-Zadeh et al. (2021) present an algorithm for this problem (a parallelization of the algorithm of Kleinberg and Leighton (2003)) that incurs at most $O(n \log \log T)$ regret. More relevantly for us, they show that this multiplicative dependence is necessary by establishing the following lower bound.

**Theorem 5.2** (Theorem 8 of Emamjomeh-Zadeh et al. (2021)). If $n = \Theta(\log \log T)$, then any deterministic algorithm for online product design must incur $\Omega(n^2/\log n)$ regret in the worst case.

We will show that it is possible to reduce the problem of online product design to the problem of contextual pricing with corruptions. In particular, we will show that any algorithm for contextual pricing with $C$ corruptions can be applied to get an equivalent regret bound in the $C$-item online product design problem.

**Lemma 5.3.** *If there exists a deterministic algorithm incurring at most $R(C,T)$ regret for the contextual pricing problem with $C$ corruptions, then there exists a deterministic algorithm incurring at most $R(C,T)$ regret for the online product design problem with $n = C + 1$ items.*

*Proof.* Consider the following algorithm for online product design (that uses the corrupted contextual pricing algorithm as a black box). At any point it will maintain a state consisting of $n + 1$ numbers: $x \in [0, 1]$, the lowest price that any item has ever sold for, and $y_1, y_2, \ldots, y_n \in [0, 1]$, where $y_i$ is the highest price that item $i$ has *not* successfully sold for. Initially, $x$ is initialized to $0$ and all $y_i$ are initialized to $1$.

The algorithm repeats the following procedure for $T$ steps:

1. Choose an item $i_t \in \arg\max_{i \in [n]} y_i$ with a maximal unsold price.

2. Ask the corrupted contextual pricing subroutine for a query price $p_t$.

3. Post price $p_t$ for item $i_t$.

4. If item $i_t$ sells at this price, update $x \leftarrow \max(x, p_t)$, and report that the item sold to the contextual pricing subroutine.

5. If item $i_t$ fails to sell at this price, update $y_{i_t} \leftarrow \min(y_{i_t}, p_t)$, and report that the item failed to sell to the contextual pricing subroutine.

We first argue that the feedback provided to the contextual pricing subroutine over the course of this process must be consistent with some valid transcript for contextual pricing with at most $C$ corruptions. In particular, let $Y$ be the maximum value of all $y_i$ at the end of the $T$ rounds, and consider the scenario where the true value of the item in the corrupted contextual pricing subroutine is equal to $Y - \epsilon$ (for an arbitrary small $\epsilon > 0$ guaranteeing $Y - \epsilon > x$). We claim this is consistent with all but at most $n - 1 = C$ observations reported to this subroutine. To see this, assume without loss of generality that $Y$ is the value of $y_1$ at the end of $T$ rounds. We claim that any other item can fail to sell at a price below $Y$ at most once – once it does so, it updates its $y_i$ to a value below $Y$, and would only be picked over $y_1$ once $y_1$ drops below $Y$ (which it never does). Since there are $n - 1 = C$ other items, there are at most $C$ such events like this, which would constitute corrupted observations reported to the contextual pricing subroutine. On the other hand, since $x$ remains below $Y$ throughout this process, we never report an error in the other direction (a sale at a price above $Y$), and hence these are the only such corruptions.

The regret of the corrupted contextual pricing subroutine under this scenario is given by

$$(Y - \epsilon)T - \sum_{t=1}^{T} \sigma_t p_t.$$

where, as before, we write $\sigma_t = 1$ if the item sells and $\sigma_t = 0$ otherwise. But now, note that the maximum value of any item in the optimal product design instance is at most $Y$, so the regret incurred by the online product design algorithm is at most

$$YT - \sum_{t=1}^{T} \sigma_t p_t.$$

It follows (by taking $\epsilon \to 0$) that any regret bound $R(T)$ for the corrupted contextual pricing subroutine extends to this algorithm for online product design. $\qquad\square$

*Proof of Theorem 5.1.* Follows immediately from Theorem 5.2 and Lemma 5.3. $\qquad\square$

# 6 One Sided Error: the Cautious Buyer

Finally, we use the ideas of the lower bound to derive an algorithm with regret $O(C + \log T)$ for what we call an "cautious buyer", which is a buyer who may not buy even if the price is below their value but at most $C$ times. The buyer never buys above their value. This corresponds to a corrupted setting with one-sided corruption: $\sigma_t \leq \mathbf{1}\{v_t \geq p_t\}$.

As in the previous section, we will state and prove this result in the non-contextual setting where the buyer's valuation can be expressed by a scalar $v^* \in [0, 1]$.

**Theorem 6.1.** There is a learning algorithm for posting a price to an cautious buyer (one-sided corrupted feedback with at most $C$ corruptions) with regret $O(C + \log T)$.

The theorem will use the algorithm by Kirkpatrick and Gao (1990) to for the following problem: consider a list of numbers $[v_1, \ldots, v_n]$, all in $[0, 1]$, unknown to the learner. The learner can at each time $t$ choose an index $i_t$ and price $p_t$ and query whether $p_{i_t} \leq v_i$. The goal of the learner is to identify an index $i$ and a price $p$ such that: $v_i \geq p \geq \max_i v_i - \epsilon$. Kirkpatrick and Gao (KG) provide an algorithm to acomplish this task with $O(n + \log(1/\epsilon))$ comparisons.

*Proof.* Apply the KG algorithm find the maximum of a list of $C+1$ numbers with parameter $\epsilon = 1/T$. For every query $p_{i_t} \leq v_i$ of the KG algorithm, respond with the result of a query $p_t \leq v^*$ to the corrupted contextual pricing algorithm. Because the buyer is cautious, the buyer may not buy a product even though the price is below $v^*$ hence the feedback for some items in the list will be consistent with a value $v_i \leq v^*$ since the KG algorithm only queries prices above the last sale and below the last no-sale for each item. For at least one of the items, however, the algorithm won't observe a corruption, so it will be able to identify the $v^*$ within a margin of $\epsilon = 1/T$ in $O(C + \log T)$ rounds. In such rounds, it will incur that much regret. From then on, the algorithm will incur at most $O(1/T)$ regret per round. $\qquad\square$

**Open Problems** We note that the lower bound in Theorem 5.2 applies to the cautious buyer as well, hence it is not possible to improve this bound to $O(C + \log \log T)$. However, we leave as an open question whether a bound of $O(C + \log T)$ is possible for a buyer with $C$ corruptions in either direction. Another interesting open direction is to provide a $O(C + d \log T)$ algorithm for a cautious buyer in the contextual setting.

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
