# OpenReview forum: "Robust Contextual Pricing"
_NeurIPS.cc/2025/Conference — NeurIPS 2025 poster_

### Official Review · Reviewer_9M6Y · 2025-06-25

**Clarity:** 4
**Significance:** 3
**Originality:** 2
**Rating:** 4
**Confidence:** 3

**Summary:**

The paper studies a version of contextual pricing robust to corruption. The buyer's valuation is represented by $\langle v*, x_t\rangle$ and the purchase decision depends on whether $v_t \geq p_t$ - the price - or not. The purchase decision represents the learner's available feedback. In the setup here, the corruption budget $C$ represents how many times the buyer can deviates from the expected purchase decision. To prove their upper-bound, that outperforms the state-of-the-art, the authors rely on $C+1$ copies of an uncorrupted contextual pricing algorithm that they update specifically in order to handle corruption. They also show that the uncorrupted algorithm proposed by Liu et al. 2021 satisfies the required assumptions and can be used as a subroutine. Finally, they extend their setup to encompass the unknown-corruption setting and prove a lower bound.

**Questions:**

I understand that the paper is mostly theoretical but do the authors know if there is empirical evidence for the use of corrupted pricing algorithm?

**Ethical Concerns:**

["NO or VERY MINOR ethics concerns only"]

**Final Justification:**

I am in favor of acceptance and hence, I maintain my score.

**Limitations:**

As said in the weaknesses, the limitation is mostly about the broad impact of the work, which appears as quite niche - although interesting - to me.

**Paper Formatting Concerns:**

Nothing special.

**Quality:**

3

**Strengths And Weaknesses:**

I believe that the main strengths are:

- the paper is very clear and easy to read.

- the proposed method based on replicata of an uncorrupted pricing algorithm to deal with the corruption is elegant and efficient.

- their method outperforms the existing bounds for this setup.

The main weaknesses are:

- the setup studied in the paper is not new and already exists, which diminishes the novelty of the work.

- the question tackled by this work is quite niche because it consists of a method that only applies to corrupted contextual pricing and very close scenarios. It is interesting but quite narrow.

---

> ### Author Rebuttal · Authors · 2025-07-31
>
> We thank the reviewer for their comments and suggestions, and we are happy that you enjoyed the paper.
>
> Re: whether there is empirical evidence for the use of corrupted pricing algorithms, online contextual pricing in the presence of noise is a practical problem arising in many different settings (where real algorithms are designed and run to address this problem). We view this work as providing a theoretical foundation for this problem in a (relatively) assumption-free / worst-case setting.  See also our response to the first reviewer.

---

### Official Review · Reviewer_8XkP · 2025-06-30

**Clarity:** 4
**Significance:** 4
**Originality:** 3
**Rating:** 5
**Confidence:** 4

**Summary:**

This paper studies the problem of robust contextual pricing. At round $t$, the seller can propose a price $p_t$, and is reported whether $v^* \cdot x_t \ge p_t$ (in which case the seller gets revenue $p_t$) or not (in which case the seller gets revenue $0$). However, there will be $C$ rounds where the feedback to the seller is corrupted.

Via a reduction to the uncorrupted case (specifically, to algorithms in the uncorrupted case that exhibit "slowdown-able" and "many-sales" properties), this paper was able to improve the following upper bounds:
- $O(C d \log \log T)$ regret in the known-corruption setting (from the SOTA $O(C d^3 \log^2 T)$).
- $O((C + \log U) d \log \log T)$ regret in the unknown-corruption-with-known-upper-bound setting (from the SOTA $O((C + \log U) d^3 \log^2 T)$).

Finally, the paper showed that any algorithm must incur a regret of $\Omega(C + d \log \log T)$, via a reduction to a lower bound for the related *online product design* problem.

**Questions:**

1) Do you think the right answer for this model would be $\Theta(C d \log \log T)$ or $\Theta(C + d \log \log T)$, or somewhere in between? In other words, do you think the $C$ multiplicative factor incurred for robustifying is necessary, or are there different ways to reduce the corrupted case to the uncorrupted case, or other techniques that one can do to mitigate this loss?

**Ethical Concerns:**

["NO or VERY MINOR ethics concerns only"]

**Final Justification:**

I have read the rebuttals and decided to keep my score.

**Limitations:**

yes

**Quality:**

4

**Strengths And Weaknesses:**

**Strengths**:
- The results are conceptually simple and clear, yet it improves the state-of-the-art by quite a large margin. Specifically, the reduction to the uncorrupted setting is easy to understand and only incurs an additional $C + 1$ loss in multiplicative factor, so one can simply verify-and-plug the state-of-the-art uncorrupted algorithm to obtain near-optimal bound.
- The paper is well-written, and the proofs provided (seem to) have no issues.

**Weaknesses**:
- A very minor weakness of this paper would be that the lower bound and upper bound is still not matching, but I would not put too much weight on this at all.

---

> ### Author Rebuttal · Authors · 2025-07-31
>
> We thank the reviewer for their encouragement and suggestions, and we are happy that you enjoyed the paper.
>
> The question you ask (of whether the multiplicative factor incurred for robustifying is necessary, or if  there are other ways to mitigate this loss) is a great question! Indeed it is one that we hope to make progress on, and address further in future works. Please note that the lower bound we establish in Section 5 provides some indication that this multiplicative factor is necessary—in particular, the lower bound says that any *deterministic* algorithm must incur the multiplicative factor, and so if we want to improve, we must look at randomized algorithms.

---

> > ### Comment · Reviewer_8XkP · 2025-08-02
> >
> > Thank you for your response!

---

### Official Review · Reviewer_NBbZ · 2025-07-01

**Clarity:** 2
**Significance:** 3
**Originality:** 2
**Rating:** 3
**Confidence:** 3

**Summary:**

The paper studies (noise-free) contextual pricing under adversarial corruptions. Specifically, in at most $C$ corrupted rounds the adversary can show to the learner an arbitrary feedback. The paper provides an algorithm with regret $O((C+1)d\log\log T)$ in the case in which $C$ is known upfront, and an algorithm with regret $O((C+\log U)d\log\log T)$ for the case in which only an upper bound $U$ on $C$ is known. The paper also provides a negative result showing that no deterministic algorithm can guarantee a regret of $O(C+\log\log T)$ in this setup (even when knowing $C$ in advance). Finally, the paper provides an algorithm guaranteeing regret $O(C+\log T)$ in the case in which the adversary is only allowed to change the feedback of no-sales to sales.

**Questions:**

can you clarify the relation between the corruption model you consider and that of Krishnamurthy et al., and comment on the precise notion of regret you’re considering?

**Ethical Concerns:**

["NO or VERY MINOR ethics concerns only"]

**Final Justification:**

Updated score to reflect clarifications from the authors.

**Limitations:**

yes

**Quality:**

2

**Strengths And Weaknesses:**

I think the paper potentially has a set of interesting results but should be improved along several directions.

First, the relation of the corruption model with those of prior works is not entirely clear. For instance, the paper repeatedly compares its results to those of Krishnamurthy et al.. However, the corruption model in Krishnamurthy et al. seems to be different from what is considered here: in their paper the adversary can manipulate the valuation, while here the adversary can only manipulate feedback. I would appreciate a more detailed discussion of the corruption model of Paes Leme et al. 2022 as well. Moreover, guarantees of Krishnamurthy et al. hold in high probability with respect to the realised losses, while from my understanding the guarantees provided in this paper hold only in expectation.

Related to the last point above, I think the paper could benefit from a rewriting in which details of the results are formally spelled out. As it is, it feels a bit handwavy in certain points (e.g., in theorem 3.1 what’s the notion of regret you’re actually bounding?).

The lower bound provided in the paper is for deterministic algorithms, but the algorithms presented are all randomised. It would be nice to have a characterisation for the same class of algorithms (or at least some discussion about that). I guess part of the problem here is using the result on online product design as a blackbox.

Section 6 is lacking many details: does the learner need knowledge of $C$ in Theorem 6.1? What is the algorithm mentioned at line 282 supposed to do? It would also be nice to add some further comments on why one sided errors make the problem fundamentally easier.

The reference for Krishnamurthy et al. should be updated since it is referring to an old version of the paper.

---

> ### Author Rebuttal · Authors · 2025-07-31
>
> We thank the reviewer for their questions and suggestions. We are glad that you found the results interesting, and we are happy to clarify some confusions, and hope that these will remove your concerns about our work, and make your evaluation more positive.
>
> **Relationship of Our Model to [KLPS20]:** The model used in our paper and that of Krishnamurthy et al. are essentially the same, despite cosmetic differences in how they appear.
> Both the models are concerned with the number of corrupted rounds, and both of them bound this number by $C$.
> In the [KLPS20] model, the adversary can manipulate the loss incurred during a corrupted time-step, whereas our adversary can corrupt the feedback. Since there are only $C$ corrupted rounds, this changes the true loss by at most O(1), since the losses are bounded.
> Therefore the difference between the losses of the two models is at most an *additive* term of $O(C)$. Since both algorithms must suffer at least $\Omega(C)$ regret, the regret is essentially the same under both models up to constant factors. We will definitely add a discussion about this to the paper.
>
> **Notion of Regret:** The notion of regret that we minimize (it is also defined in line 105) is to minimize the sum of the losses. We will edit the paper to make this more clear, and also work on the rest of the paper to make the presentation more transparent.
>
> **Randomized vs Deterministic Algorithms:** Understanding what is possible in the corrupted setting with fully deterministic algorithms (or conversely, proving lower bounds that apply to randomized algorithms) is an interesting future direction. Despite this mismatch, we still feel that the lower bound we establish in Section 5 is valuable by providing some evidence that a multiplicative $C$ factor is required.
>
> **Details in Section 6:** Thanks for pointing this out, we will definitely clarify the setting of Section 6. In that section, the learner does know the total number of corruptions $C$ (or at least, an upper bound on this). Moreover, the Kirkpatrick-Gao algorithm mentioned in line 282 solves the following problem: find the maximum of $C$ real numbers in [0, 1] to within precision $\epsilon$ using only $C + \log(1/\epsilon)$ threshold queries (of the form, “is $x_i > v$” for given $i \in [C]$ and $v \in [0,1]$). We will reword and emphasize this in the revised version.

---

> > ### Comment · Reviewer_NBbZ · 2025-08-04
> >
> > Thank you for the clarifications. I remain unconvinced by the insights we can get from the lower bound for the class of algorithms you consider. I’m fine with the remaining explanations and will adjust my score accordingly.

---

### Official Review · Reviewer_uZ1C · 2025-07-02

**Clarity:** 4
**Significance:** 2
**Originality:** 3
**Rating:** 4
**Confidence:** 5

**Summary:**

The paper studies contextual pricing under adversarially corrupted binary feedback.
In each round \\( t \\in [T] \\), a seller observes a feature vector \\( x_t \\in \\mathbb{R}^d \\) that describes the product, posts a price \\( p_t \\in [0,1] \\), and observes only whether the buyer purchases (\\( \\sigma_t = 1 \\)) or not (\\( \\sigma_t = 0 \\)).
The buyer’s latent valuation is assumed linear: \\( v_t = \\langle v^\\star, x_t \\rangle \\), but the feedback can be arbitrarily corrupted in at most \\( C \\) rounds.
The goal is to minimize regret — the cumulative revenue loss relative to the clairvoyant seller.

**Known-\\( C \\)** case.
The authors give a reduction that calls any uncorrupted contextual pricing algorithm as a black box.
Instantiating it with the best current algorithm of Liu et al. (2021) yields a regret bound:
\\[
O\\left( (C+1)\\, d \\log \\log T \\right),
\\]
improving the previous
\\[
O\\left( C\\, d^3 \\log^2 T \\right)
\\]
bound of Krishnamurthy et al. (2020).

**Unknown-\\( C \\)** (upper bound \\( U \\) known).
Using a multi-layer update-probability schedule (à la Lykouris et al., 2018), they obtain:
\\[
O\\left( (C + \\log U)\\, d \\log \\log T \\right)
\\]
with high probability.

For the non-contextual case (\\( d = 1 \\)), they prove that no deterministic algorithm can achieve \\( O(C + d \\log \\log T) \\) regret.
When \\( C = \\Theta(\\log \\log T) \\), any algorithm incurs at least
\\[
\\Omega\\left( \\frac{C^2}{\\log C} \right),
\\]
demonstrating that robustifying pricing is inherently harder than robustifying \\( \\varepsilon \\)-ball contextual search.

For a cautious buyer — who may refuse bargains but never overpays — they design an
\\( O(C + \\log T) \\) algorithm in the non-contextual case, matching their lower-bound order (up to \\( \\log \\log T \\) vs. \\( \\log T \\)).

**Methodology**
The key idea is to run \\( C + 1 \\) (or \\( \\lceil \\log U \\rceil \\)) parallel copies of a standard uncorrupted learner (e.g., the Liu–Leme–Schneider 2021 algorithm with \\( O(d \\log \\log T) \\) regret).

1. On each context \\( x_t \\), all copies propose prices; the maximum is posted.
   This ensures that a sale implies every other copy’s price was also acceptable, allowing one to charge the loss of the master algorithm to (essentially) the best uncorrupted copy.

2. Update rules are randomized so that, in expectation, at least one copy receives no corrupted feedback, yet every copy is slowed down by at most a factor \\( O(C) \\) (or \\( O(C / \\delta) \\) in the high-probability version).

A potential-function analysis borrowed from Liu et al. (2021) then bounds the regret of each slowed-down copy, which transfers to the master algorithm.

The lower bound reduces online product design with \\( n \\approx C \\) items (Emamjomeh-Zadeh et al., 2021) to contextual pricing with \\( C \\) corruptions, showing that an additive \\( C \\) price for robustness is unachievable.

**Questions:**

Same as weakness

**Ethical Concerns:**

["NO or VERY MINOR ethics concerns only"]

**Final Justification:**

Dear AC and authors,

After reading all the comments and responses, I am leaning slightly toward acceptance, given the contribution in tightening the upper bound. My concern about the true meaningfulness of the model remains, but I feel this should not detract from its contribution to the literature.

Best wishes,
Your reviewer

**Limitations:**

Same as weakness

**Paper Formatting Concerns:**

No concern

**Quality:**

3

**Strengths And Weaknesses:**

**Strengths**
The paper offers a clean and modular algorithmic framework that black-boxes any uncorrupted learner, leading to easy implementation and broad applicability.
It significantly improves regret bounds by reducing their dependence on both dimension \\( d \\) and corruption level \\( C \\), and it sharpens theoretical understanding through matching lower bounds.
The black-box reduction paradigm has potential to inspire similar techniques in other contextual decision problems with binary feedback.

**Limitation**
I greatly appreciate the paper’s clear presentation and its solid theoretical foundation. My main reservation concerns the model’s practical applicability. Could you please provide concrete examples or application domains in which this type of pricing problem naturally arises?

In addition, it would be very helpful to include numerical experiments that benchmark the proposed algorithms against established methods in the literature. Such results would illustrate the empirical effectiveness of your approach and strengthen the paper’s practical contribution.

---

> ### Author Rebuttal · Authors · 2025-07-31
>
> We thank the reviewer for their encouragement and suggestions. We are happy to hear that you liked the modular and conceptually clean framework for robust learning; indeed, we too feel this has broad potential. Below, please find responses to your comments.
>
> **Empirical Results and Relevance of our algorithm:** Although the focus of our paper was purely theoretical, the problem itself is well-motivated by practical considerations. In many settings when sellers are offering a set of differentiated products to a variety of users, they can determine the price of each item by trying to understand the market value for each of its intrinsic features. For example, [LPLV18] and [CLPL19] both use AirBnb or online advertising as examples of such a phenomenon. We will definitely consider empirical validation in our future work.

---

> > ### Comment · Reviewer_uZ1C · 2025-08-03
> > **Real relevance of algorithms**
> >
> > Thank you for answering my question. However, your reply left me a bit more confused.
> >
> > You mentioned: "When sellers are offering a set of differentiated products to a variety of users, they can determine the price of each item by trying to understand the market value for each of its intrinsic features." I agree with this. However, in that case, wouldn’t the customer be choosing one or more products from a set, rather than facing a single-offer and threshold decision, as modeled in the paper?
> >
> > On the Airbnb example, users typically choose one option from a set of listings, which seems different from the model studied in the paper.
> >
> > Also, could you please provide more details about the papers you mentioned? I couldn’t identify them from your response.
> >
> > Thank you!

---

### Decision · Program_Chairs · 2025-09-17

**Decision:**

Accept (poster)

**Comment:**

The paper studies contextual pricing with linear valuations that might be corrupted. At each round in the uncorrupted case, the learner observes a $d$-dimensional feature vector $x_t$ for the item being sold, and the buyer's valuation is $x_t^T v^\*$ for some unknown $v^\*$. The learner sets a price, and the buyer decides to purchase the item if their valuation exceeds the price. The learner only receives feedback on whether a purchase occurs and aims to set prices that maximize its cumulative revenue. In the corrupted case, an adversary can alter the feedback observed by the learner in up to $C$ rounds. The authors prove an upper bound of $O(Cd\log\log T)$ when $C$ is known and $O((C+\log U)d\log\log T)$ if $U$ is an upper bound on $C$, thus improving the existing upper bound of $O(Cd^3\log^2 T)$. They also prove a lower bound for deterministic algorithms, showing that no algorithm can guarantee a bound of $O(C+\log\log T)$. Finally, they provide an algorithm that achieves a regret of  $O(C+\log T)$ when the adversary can only switch no-sales to sales.

The reviewers all appreciated the proposed algorithmic approach and the improvement in the upper bounds compared to existing algorithms. The main concern raised by the reviewers regards the lower bound - both due to the gap from the upper bound and the fact that it only holds for deterministic algorithms (whereas the algorithm in the paper is randomized). Despite these concerns, the overall contributions, particularly the improved upper bounds and the establishment of the first lower bound for this problem, led me to recommend accepting the paper.

The reviewers also identified several issues related to clarity and the connection of this model to prior work. The authors are asked to revise the text to address these points in accordance with their rebuttal.